# The Information Geometry of Sensor Configuration

**DOI:** 10.3390/s21165265

**Published:** 2021-08-04

**Authors:** Simon Williams, Arthur George Suvorov, Zengfu Wang, Bill Moran

**Affiliations:** 1Department of Electrical and Electronic Engineering, University of Melbourne, Melbourne, VIC 3000, Australia; wmoran@unimelb.edu.au; 2Department of Theoretical Astrophysics, Eberhard Karls University of Tübingen, D-72076 Tübingen, Germany; suvorovarthur@gmail.com; 3Manly Astrophysics, 15/41-42 East Esplanade, Manly, NSW 2095, Australia; 4School of Automation, Northwestern Polytechnical University, Xi’an 710072, China; wangzengfu@gmail.com

**Keywords:** information geometry, sensor management, riemannian geometry

## Abstract

In problems of parameter estimation from sensor data, the Fisher information provides a measure of the performance of the sensor; effectively, in an infinitesimal sense, how much information about the parameters can be obtained from the measurements. From the geometric viewpoint, it is a Riemannian metric on the manifold of parameters of the observed system. In this paper, we consider the case of parameterized sensors and answer the question, “How best to reconfigure a sensor (vary the parameters of the sensor) to optimize the information collected?” A change in the sensor parameters results in a corresponding change to the metric. We show that the change in information due to reconfiguration exactly corresponds to the natural metric on the infinite-dimensional space of Riemannian metrics on the parameter manifold, restricted to finite-dimensional sub-manifold determined by the sensor parameters. The distance measure on this configuration manifold is shown to provide optimal, dynamic sensor reconfiguration based on an information criterion. Geodesics on the configuration manifold are shown to optimize the information gain but only if the change is made at a certain rate. An example of configuring two bearings-only sensors to optimally locate a target is developed in detail to illustrate the mathematical machinery, with Fast Marching methods employed to efficiently calculate the geodesics and illustrate the practicality of using this approach.

## 1. Introduction

This paper is an attempt to initiate the construction of an abstract theory of sensor management, in the hope that it will help to provide both a theoretical underpinning for the solution of practical problems and insights for future work. A key component of sensor management is the amount of information a sensor in a given configuration can gain from a measurement, and how that information gain changes with the configuration of the sensor. In this vein, it is interesting to observe how information theoretic *surrogates* have been used in a range of applications as objective functions for sensor management; see, for instance, [1,2,3]. Our aim here is to abstract from various papers including these, those by Kershaw and Evans [4] on sensor management, as well as others, the mathematical principles required for this theory. Our approach is set within the mathematical context of differential geometry and we aim to show that geodesics on a particular manifold provide the theoretical best-possible sensor reconfiguration in the same way that the Cramer–Rao lower bound provides a minimum variance for an estimator.

The problem of estimation is expressed in terms of a likelihood; that is, a probability density p(x|θ), where *x* denotes the measurement and θ the parameter to be estimated. It is well known [5] that the Fisher information associated with this likelihood provides a measure of information gained from the measurement. This in turn leads to the Fisher–Rao metric [6,7] on the parameter space. This Riemannian metric lies at the heart of our discussion.

While the concepts discussed here are, in some sense, generic and can be applied to any sensor system that has the capability to modify its characteristics, for simplicity and to keep in mind a motivating example, we focus on a particular problem: that of localization of a *target*. Of course, a sensor system is itself just a (more complex) aggregate sensor, but it will be convenient, for the particular problems we will discuss, to assume a discrete collection of disparate (at least in terms of location) sensors, that together provide measurements of aspects of the location of the target. This distributed collection of sensors, each drawing measurements that provide partial information about the location of a target, using known likelihoods, defines a particular *sensor configuration state*. As the individual sensors move, they change their sensing characteristics and thereby the collective Fisher information associated with estimation of target location. As already stated, the Fisher information matrix defines a metric, the Fisher–Rao metric, over the physical space where the target resides, or in more generality a metric over the parameter space in which the estimation process takes place, and this metric is a function of the location of the sensors. This observation permits analysis of the information content of the system, as a function of sensor parameters, in the framework of differential geometry (“information geometry”) [8,9,10,11]. A considerable literature is dedicated to the problem of optimizing the configuration so as to maximize information retrieval [1,12,13,14]. The mathematical machinery of information geometry has led to advances in several signal processing problems, such as blind source separation [15], gravitational wave parameter estimation [16], and dimensionality reduction for image retrieval [17] or shape analysis [18].

In sensor management/adaptivity applications, the performance of the sensor configuration (in terms of some function of the Fisher information) becomes a cost associated with finding the optimal sensor configuration, and tuning the metric by changing the configuration serves to optimize this cost. Literally hundreds of papers, going back to the seminal work of [4] and perhaps beyond, have used the Fisher information as a measure of sensor performance. In this context, parametrized families of Fisher–Rao metrics arise (e.g., [19,20]). Sensor management then becomes one of choosing an optimal metric (based on the value of the estimated parameter), from among a family of such, to permit the acquisition of the maximum amount of information about that parameter.

As we have stated, the focus, hopefully clarifying, example of this paper, is that of estimating the location of a target using measurements from mobile sensors (cf. [21,22]). The information content of the system depends both on the location of the target and on the spatial locations of the sensors, because the covariance of measurements is sensitive to the distances and angles made between the sensors and the target. As the sensors move in space, the associated likelihoods vary, as do the resulting Fisher matrices, describing the information content of the system, for every possible sensor arrangement. It is this interaction between sensors and target that this paper sets out to elucidate in from an information geometric viewpoint.

Perhaps the key extra idea in this paper is the observation by Gil-Medrano and Michor that the collection of all Riemannian metrics on a Riemannian manifold itself admits the structure of an infinite-dimensional Riemannian manifold [23,24]. Of interest to us is only the subset of Riemannian metrics corresponding to Fisher informations of available sensor configurations, and this allows us to restrict attention to a finite-dimensional sub-manifold of the manifold of metrics, called the *sensor manifold* [12,13]. In particular, a continuous adjustment of the sensor configuration, say by moving one of the individual sensors, results in a continuous change in the associated Fisher metric and so a movement in the sensor manifold.

Though computationally difficult, the idea of regarding the Fisher–Rao metric as a measure of performance of a given sensor configuration and then understanding variation in sensor configuration in terms of the manifold of such metrics is a powerful concept. It permits questions concerning optimal target trajectories, as discussed here, to minimize information passed to the sensors and, as will be discussed in a subsequent paper, optimal sensor trajectories to maximize information gleaned about the target. In particular, we remark that the metric on the space of Riemannian metrics that appears naturally in a mathematical context in [23,24] also has a natural interpretation in a statistical context.

Our aims here are to further develop the information geometric view of sensor configuration begun in [12,13]. While the systems discussed are simple and narrow in focus, already they point to concepts of information collection that appear to be new. Specifically, we setup the target location problem in an information geometric context and show that the optimal (in a manner to be made precise in Section 3) sensor trajectories, in a physical sense, are determined by solving the geodesic equations on the configuration manifold (Section 4). Various properties of geodesics on this space are derived, and the mathematical machinery is demonstrated using concrete physical examples (Section 5).

## 2. Divergences and Metrics

Here, we discuss some generalities on statistical divergences and metrics. The two divergences of interest to us are the Kullback–Leibler divergence and the Jensen–Shannon divergence. These are defined by
(1)$$\begin{array}{cc} D_{\mathrm{KL}}\left(p∥q\right) & =\int p\left(x\right)\log\left(\frac{p(x)}{q(x)}\right)\;dx \end{array}$$
(2)$$\begin{array}{cc} D_{\mathrm{JS}}\left(p∥q\right) & =\frac{1}{2}\left[D_{\mathrm{KL}}\left(p∥\frac{p+q}{2}\right)+D_{\mathrm{KL}}\left(q∥\frac{p+q}{2}\right)\right], \end{array}$$
where *p* and *q* are two probability densities on some underlying measure space.

In our case, the densities p,q are taken from a single parametrized family $p_{(}\theta )$, where θ is in a manifold *M*. More specifically, the parameter θ=θ, the location of the target. A justification for the Kullback–Leibler divergence is that, in this context, given a measurement x of a target at θ, the likelihood that the same measurement could be obtained from a target at $\theta^{\prime }$, $L(\theta ,\theta^{\prime })$, is given by the log odds expression
$$L\left(\theta ,\theta^{\prime }\right)=\log\frac{p(x|\theta )}{p(x|\theta^{\prime })},$$
and the average over all measurements is, by definition, the Kullback–Leibler divergence [25], $D_{\mathrm{KL}}\left(\theta ∥\theta^{\prime }\right)$.

Each of the Kullback–Leibler and Jensen–Shannon divergences is regarded as a measure of information difference between the two densities. They are both non-negative and share the property that D(p∥q)=0 implies that p=q almost everywhere. The Kullback–Leibler divergence is related to mutual information, as is, somewhat more circuitously the Jensen–Shannon divergence. We refer the reader to Section 17.1 of [5] for a discussion of this connection for the Kullback–Leibler, and to [26] for a brief discussion of the Jensen–Shannon divergence.

Importantly for us, as $\theta^{\prime }\to \theta$, the first non-zero term in the series expansion is second order, viz.
(3)$$\lim_{\theta^{\prime }\to \theta }D(\theta ∥\theta^{\prime })=\lim_{\theta^{\prime }\to \theta }{\left(\theta -\theta^{\prime }\right)}^{T}g\left(\theta -\theta^{\prime }\right)+O\left({\left(\theta -\theta^{\prime }\right)}^{3}\right),$$
where g=g(θ) is an n×n symmetric matrix, and *n* is the dimension of the manifold *M*.

This location-dependent matrix defines a Riemannian metric over *M*, the *Fisher information metric* [6,7]. It can also be calculated, under mild conditions, as the expectation of the tensor product of the gradients of the log-likelihood ℓ=logp(x|θ) as
(4)$$g=g\left(\theta \right)=E_{x|\theta }\left[d_{\theta }\ell ⊗d_{\theta }\ell \right].$$

This is an example of a more general structure encapsulated by the following:

**Theorem** **1.**
*Given a likelihood ℓ:M→[0,1] and a divergence $D(\ell \left(\theta \right)∥\ell \left(\theta^{\prime }\right))$ then there is a unique largest metric d on M such that $d\left(\theta ,\theta^{\prime }\right)\leq \sqrt{D(\ell \left(\theta \right)||\ell \left(\theta^{\prime }\right))}$*


The relationship between this metric and information is explored in the next section. A proof of the theorem appears in the Appendix A.

## 3. The Information in Sensor Measurements

Sensor measurements, as considered in this paper, can be formulated as follows. Suppose we have, in a fixed manifold *M*, a collection of *N* sensors located at $\lambda_{i}$ (i=1,…,N). For instance, the manifold may be $R^{3}$ and location may just mean that in the usual sense in Euclidean space. The measurements from these sensors are used to estimate the location of a target θ also in *M* (see the left of Figure 1). Each sensor draws measurements $x_{i}$ from a distribution with probability density function (PDF) $p_{i}\left(x_{i}|\theta ,\nu_{i}\right)$, where θ represents the state of the system we are trying to estimate and $\nu_{i}$ the noise parameters of each individual sensor.

A measurement $x={\left\{x_{i}\right\}}_{i=1}^{N}$ is the collected set of individual measurement from each of the sensors with likelihood
(5)$$p\left(x|\theta \right)=\prod_{i=1}^{N}p_{i}\left(x_{i}|\theta ,\nu_{i}\right).$$These measurements are assumed independent between sensors and over time when dynamics are included. While this assumption is probably not necessary, it allows one to define the aggregate likelihood (Equation 5) as a simple product over the individual likelihoods, which renders the problem computationally more tractable.

Since Fisher information is additive over independent measurements, the Fisher information metric provides a measure of the instantaneous change in information the sensors can obtain about the target. In this paper, we adopt a relatively simplistic view that the continuous collection of measurements as exemplified by, for instance, the Kalman-Bucy filter [27], is a limit of measurements discretized over time. Because sensor measurements depend on the relative locations of the sensors and target, this incremental change depends on the direction the target is moving; the Fisher metric (Equation 4) can naturally be expressed in coordinates that represent the sensor locations (see Section 4), but also depends on parameters that represent target location, which may be functions of time in a dynamical situation. Once the Fisher metric (Equation 4) has been evaluated, one can proceed to optimize it in an appropriate manner.

Another divergence will be of interest in our discussions. This is the Jensen–Shannon divergence, defined in terms of the Kullback–Leibler divergence by
(6)$$D_{\mathrm{JS}}\left(p_{\theta }∥p_{\theta^{\prime }}\right)=\frac{1}{2}\left[D_{\mathrm{KL}}\left(p_{\theta }∥\frac{p_{\theta }+p_{\theta^{\prime }}}{2}\right)+D_{\mathrm{KL}}\left(p_{\theta^{\prime }}∥\frac{p_{\theta }+p_{\theta^{\prime }}}{2}\right)\right]$$

### 3.1. D-Optimality

The information of the sensor system described in Section 3 is a matrix-valued function and so problems of optimization of ‘total information’ with respect to the sensor parameters are not well defined. We require a definition of ‘optimal’ in an information theoretic context. Several different optimization criteria exist (e.g., [28,29]), achieved by constructing scalar invariants from the matrix entries of *g*, and maximizing those functions in the usual way.

We adopt the notion of *D-optimality* in this paper; we consider the maximization of the determinant of (Equation 4). Equivalently, D-optimality maximizes the differential Shannon entropy of the system with respect to the sensor parameters [30], and minimizes the volume of the elliptical confidence regions for the sensors estimate of the location of the target θ [31].

While we have focused so far on the metric as a function of target location θ on the manifold, it is also clear that sensor parameters are involved. In applying D-optimality (or any other) criterion to this problem, we take account of the possibility that sensor locations (in the context we have described) and distributions are not fixed. Conventionally, measurements are drawn from sensors with *fixed* properties, with a view to estimating a parameter θ. Permitting sensors to move throughout *M* produces an infinite family of sensor configurations, and hence Fisher–Rao metrics (Equation 4), parametrized by the locations of the sensors. One aim of this structure is to move sensors to locations that serve to maximize information content, given some prior distribution for a particular θ∈M. This necessitates a tool to measure the difference between information provided by members of a family of Fisher–Rao metrics; this is explored in Section 4. We emphasize that the ideas described here will extend to situations where the sensor manifold is not identified with a copy (or several copies) of the target manifold.

### 3.2. Geodesics on the Sensor Manifold

We now consider the case where the target is in motion, so that θ varies along a path γ(t)⊂M. The instantaneous information gain by the sensor(s) at time *t* is then $g\left(\gamma^{\prime }\left(t\right),\gamma^{\prime }\left(t\right)\right)$, where *g* is the Fisher information metric (Equation 4). We define the information provided to the sensors about the target by the measurements as the target moves along the curve as follows. Take a δ-partition $T=\{0=t_{0},t_{1},t_{2},\ldots ,t_{N}=1\}$ of the interval [0,1], where $0<t_{i+1}-t_{i}<\delta$ (i=0,1,…,N−1). At each point $\gamma (t_{i})$, a measurement is taken represented by the likelihood $p(x_{i}|\gamma \left(t_{i}\right))$. Over the interval $\left(t_{i+1},t_{i}\right]$, the increment in information could be $D(\gamma (t_{i}∥\gamma \left(t_{i+1}\right)/\left(t_{i+1}-t_{i}\right)$ but this does not make sense for an increment as it is not symmetric. Instead, we should calculate the information gain from the endpoints of the interval to the mean distribution
$$D(\gamma (t_{i}∥\frac{1}{2}\left(\gamma \left(t_{i}\right)+\gamma \left(t_{i+1}\right)\right)+D(\gamma (t_{i+1}∥\frac{1}{2}\left(\gamma \left(t_{i}\right)+\gamma \left(t_{i+1}\right)\right).$$This is also know as the Jensen–Shannon divergence, $D_{JS}\left(X∥Y\right)$. Based on the assumption that measurements taken over time are all independent, as the target traverses a curve γ, the total information $I_{T}$ gained measuring at the δ-partition, *T* is
(7)$$I_{T}\left(\gamma \right)=\sum_{i=0}^{N-1}D_{JS}(\gamma (t_{i}∥\gamma \left(t_{i+1}\right)/\left(t_{i+1}-t_{i}\right)$$Letting δ go to zero gives us an information functional, I(T), defined on paths, γ,
(8)$$I\left(\gamma \right)=\lim_{\delta \to 0}I_{T}\left(\gamma \right)=\int_{0}^{T}g\left(\gamma^{\prime }\left(t\right),\gamma^{\prime }\left(t\right)\right)\;dt,$$
which is the equivalent of the energy functional in differential geometry (e.g., Chapter 9 of [32]) This has the same extremal paths as $l_{g}\left(\gamma \right)$, the arc-length of the path γ,
(9)$$l_{g}\left(\gamma \right)=\int_{0}^{T}\sqrt{g\left(\gamma^{\prime }\left(t\right),\gamma^{\prime }\left(t\right)\right)}\;dt.$$

Paths with extremal values of this length are *geodesics* and these can be interpreted as the evasive action that can be taken by the target to minimize amount of information it gives to the sensors.

### 3.3. Kinematic Conditions on Information Management

While the curves that are extrema of the information functional and of arc-length are the same as sets, a geodesic only minimizes the information functional if traversed at a specific speed, namely,
(10)$$dl_{g}/dt=+\sqrt{g(\gamma^{\prime }\left(t\right),\gamma^{\prime }\left(t\right))}.$$

In differential geometric terms, this is equivalent to requiring the arc-length parameterization of the geodesic to fulfill the energy condition. In order to minimize information about its location, the target should move along a geodesic of *g* at exactly the speed $dl_{g}/dt$ (Equation 10). This quantifies directly the relation between the speed of the target and the ability to identify its location.

While aspects of this speed constraint are still unclear to us, an analogy that may be useful is to regard the target as moving though a “tensorial information fluid” whose flow is given by the information metric. In this setting, moving slower relative to the fluid will result in “pressure” building behind, requiring information (energy) to be expended to maintain the slower speed. Moving faster also requires more information to push through the slower moving fluid. This information is directly available to the sensors resulting in improved target position estimate.

In the fluid dynamics analogy, the energy expended in moving though a fluid is proportional to the square of the difference in speed between the fluid and the object. The local energy is proportional to the difference between actual speed and the speed desired by the geodesic; that is, the speed that minimizes the energy functional. Pursuing the fluid dynamics analogy, the density of the fluid mediates the relationship between the energy and the relative speed.
E∝g(δv,δv)In particular, the scalar curvature, which depends on *G*, influences the energy and hence the information flow. We will explore this issue further in a future publication.

## 4. The Information of Sensor Configurations

A sensor configuration is a set $\Gamma ={\left\{\lambda_{i}\right\}}_{i=1}^{N}$ of sensor parameters. The Fisher–Rao metric *g* can be viewed as a function of Γ as well as θ, the location of the target. To calculate the likelihood that a measurement came from one configuration $\Gamma_{0}$ over another $\Gamma_{1}$ requires the calculation of $p\left(x|\Gamma_{0},\theta \right)/p\left(x|\Gamma_{1},\theta \right)$, which is difficult as the value of θ is not known exactly. Measurements can be used to construct an estimate $\hat{\theta }$. However, the distribution of this estimate is hard to quantify and even harder to calculate. Instead, here, the maximum entropy distribution is used. This is normally distributed with mean $\hat{\theta }$, and covariance $g^{-1}\left(\hat{\theta }\right)$, the inverse of the Fisher information metric at the estimated target location.

The information gain due to the sensor configuration Γ is now $D(p\left(x|\Gamma ,\hat{\theta }\right)∥1)$ because there was no prior information about the location before the sensors were configured compared with the maximum entropy distribution *p* after. Note that the uniform distribution, 1 is, in general, an improper prior in the Bayesian sense unless the manifold *M* is of finite volume. It may be necessary, therefore, to restrict attention to some (compact) submanifold Ω⊂M for $D(p\left(x|\Gamma ,\hat{\theta }\right)∥1)$ to be well defined (cf. the discussion below Equation (Equation 12)).

Evaluating this information gain gives
(11)$$D\left(p∥1\right)=\log\left({\left(2\pi e\right)}^{n}\det g^{-1}\left(\Gamma ,\hat{\theta }\right)\right)$$

The Fisher information metric *G* for this divergence can be calculated from
(12)$$G\left(h,k\right)=E\left[d_{g}^{2}D\left(p∥1\right)\right]=\int_{M}tr\left(g^{-1}hg^{-1}k\right)\mathrm{vol}\left(g\right)\;d\mu$$
where *h* and *k* are tangent vectors to the space of metrics. The integral defining (Equation 12) may not converge for non-compact *M*, so restriction to a compact submanifold Ω of *M* is assumed throughout as necessary (cf. Figure 1).

### 4.1. The Manifold of Riemannian Metrics

The set of all Riemannian metrics over a manifold *M* can itself be imbued with the structure of an infinite-dimensional Riemannian manifold [23,24], which we call M. Points of M are Riemmanian metrics on *M*; i.e., each point G∈M bijectively corresponds to a positive-definite, symmetric (0,2)-tensor in the space $S_{+}^{2}T^{☆}M$. Under reasonable assumptions, an $L^{2}$ metric on M [23,33] may be defined as:(13)$$G\left(h,k\right)=\int_{M}tr\left(g^{-1}hg^{-1}k\right)\mathrm{vol}\left(g\right)\;d\mu ,$$
which should be compared to (Equation 12).

It should be noted that the points of the manifold M comprise *all* of the metrics that can be put on *M*, most of which are irrelevant for our physical sensor management problem. We restrict consideration to a sub-manifold of M consisting only of those Riemannian metrics that are members of the family of Fisher information matrices (Equation 4) corresponding to feasible sensor configurations. This particular sub-manifold is called the ‘sensor’ or ‘configuration’ manifold [12,13] and is denoted by M(Γ), where now the objects *h* and *k* are now elements of the now finite-dimensional tangent space TM(Γ). The dimension of M(Γ) is N×dim(M) since each point of M(Γ) is uniquely described by the locations of the *N*-sensors, each of which require dim(M) numbers to denote their coordinates. A visual description of these spaces is given in Figure 1. For all cases considered in this paper, the integral defined in (Equation 13) is well defined and converges (see, however, the discussion in [21]).

For the purposes of computation, it is convenient to have an expression for the metric tensor components of (Equation 13) in some local coordinate system. In particular, in a given coordinate basis $z^{i}$ over M(Γ) (not to be confused with the coordinates on Ω; see Secion Section 4), the metric (Equation 13) reads
(14)$$G\left(h,k\right)=\int_{\Omega }g^{nk}g^{\ell m}h_{mn}k_{\ell k}\mathrm{vol}\left(g\right).$$
where *h* and *k* are tangents vectors in TM(Γ) given in coordinates by
(15)$$TM\left(\Gamma \right)=\mathrm{span}{\left\{\frac{\partial }{\partial z^{i}}g_{mn}\right\}}_{i=1}^{\dim M(\Gamma )}.$$From the explicit construction (Equation 14), all curvature quantities of M(Γ), such as the Riemann tensor and Christoffel symbols, can be computed.

### 4.2. D-Optimal Configurations

D-optimality in the context of the sensor manifold described above is discussed in this section. Suppose that the sensors are arranged in some arbitrary configuration $\Gamma_{0}$. The sensors now move in anticipation of target behaviour; a prior distribution is adopted to localize a target position θ. The sensors move continuously to a new configuration $\Gamma_{1}$, where $\Gamma_{1}$ is determined by maximizing the determinant of *G*, i.e., $\Gamma_{1}$ corresponds to the sensor locations for which det(G), computed from (Equation 14), is maximized. The physical situation is depicted graphically in Figure 2. This process can naturally be extended to the case where real measurement data are used. In particular, as measurements are drawn, a (continuously updated) posterior distribution for θ becomes available, and this can be used to update the Fisher metric (and hence the metric *G*) to define a sequence $\Gamma_{t}$ of optimal configurations; see Section 5.

### 4.3. Geodesics for the Configuration Manifold

While D-optimality allows us to determine *where* the sensors should move given some prior, it provides us with no guidance on which *path* the sensors should traverse, as they move through Ω, to reach their new, optimized positions.

A path Υ(t) from one configuration $\Gamma_{0}$ to another $\Gamma_{1}$ is a set of paths $\Gamma \left(t\right)={\left\{\gamma_{i}\left(t\right)\right\}}_{i=1}^{N}$ for each sensor $S^{i}$ from location $\lambda_{i}^{0}$ to $\lambda_{i}^{1}$. Varying the sensor locations is equivalent to varying the metric $g\left(t\right)=g(\Gamma \left(t\right),\hat{\theta }\left(t\right))$ and the estimate of the target location $\hat{\theta }$. The information gain along Υ, I(Υ), is then
(16)$$I\left(\Upsilon \right)=\int_{\Upsilon }G_{g(t)}\left(g^{\prime }\left(t\right),g^{\prime }\left(t\right)\right)\;dt,$$
and the extremal paths are again the geodesics of the metric *G*, by analogy with (Equation 8). Also the speed constraint observed earlier in Section 3. Section 3.3 for the sensor geodesics is in place here and given by
(17)$$\frac{dl_{G}}{dt}=\sqrt{G_{g(t)}\left(g^{\prime }\left(t\right),g^{\prime }\left(t\right)\right)}.$$

Again, this leads to the conclusion that there are kinematic constraints on the rate of change of sensor parameters that lead to the collection of the maximum amount of information. In this case, it is the sensor using information to slow or increase the rate of change of its parameters relative to (Equation 17). That information is wasted (like a form of information heat) and results in less information available for localisation of the target and, hence, a poorer estimate of it location.

## 5. Configuration for Bearings-Only Sensors

To illustrate the mathematical machinery developed in the previous sections consider first the configuration metric for two bearings-only sensors. The physical space Ω, where the sensors and the target reside, is chosen to be the square $\left[-10,10\right]\times \left[-10,10\right]\subset R^{2}$, though the sensors are ‘elevated’ at some height $z_{i}$ above the plane. Effectively, we assume an uninformative (uniform) prior over the square Ω.

The goal is to estimate a position θ from bearings-only measurements taken by the sensors, as in previous work [12,13]. We assume that measurements are drawn from a Von Mises distribution,
(18)$$M_{n}∼p_{n}\left(\cdot |\theta \right)=\frac{e^{\kappa \cos\left[\cdot -\arg\left(\theta -\lambda_{n}\right)\right]}}{2\pi I_{0}\left(\kappa \right)},$$
where κ is the concentration parameter and $I_{r}$ is the *r*th modifed Bessel function of the first kind, [34], and $\lambda_{n}=\left(x_{n},y_{n}\right)$ is the location of the *n*-th sensor in Cartesian coordinates. Note that the parameter, κ, being a measure of angular concentration means the location error will increase and the signal-to-noise ratio decrease as the sensors move farther away from the target.

For the choice (Equation 18), the Fisher metric (Equation 4) can be computed easily, and has components
(19)$$\begin{array}{cc} g= & \kappa \left(1-\frac{I_{2}\left(\kappa \right)}{2I_{0}\left(\kappa \right)}\right)\\ & \sum_{i=1}^{N}\frac{1}{{\left(x-x_{i}\right)}^{2}+{\left(y-y_{i}\right)}^{2}+{\left(z-z_{i}\right)}^{2}}\left(\begin{array}{cc} {\left(y-y_{i}\right)}^{2} & -\left(x-x_{i}\right)\left(y-y_{i}\right)\\ -\left(x-x_{i}\right)\left(y-y_{i}\right) & {\left(x-x_{i}\right)}^{2} \end{array}\right), \end{array}$$
where κ=1 is chosen as our noise parameter in what follows.

The target lives in the plane, thus z=0 but the sensors live in a copy of the plane at height 1, i.e., we take $z_{i}=1$. This prevents singularities from emerging within (Equation 19).

### 5.1. Target Geodesics

A geodesic, γ(t), starting at p with initial direction v for a manifold with metric *g* is the solution of the coupled second-order initial value problem for the components $\gamma^{i}\left(t\right)$:(20)$$\frac{d^{2}\gamma^{i}}{dt^{2}}=\Gamma_{jk}^{i}\frac{d\gamma^{j}}{dt}\frac{d\gamma^{k}}{dt},\;\gamma \left(0\right)=p,\;\gamma^{\prime }\left(0\right)=v,$$
where $\Gamma_{jk}^{i}$ are the Christoffel symbols for the metric *g*.

Figure 3 shows directly integrated solutions to the geodesic Equation (Equation 20) a target at (−1,−3) and sensors at (−7,−6) and (0,1) (raised at a height 1 above Ω). The differing paths correspond to the initial direction vector (cosϕ,sinϕ) of the target, varying as ϕ varies from 0 to 2π radians in steps of 0.25 radians.

The solution of the geodesic equation are generated from t=0 to 5. this means the length of the corresponds to the information given away by the target. The longest line corresponds to the target forming a right-angle with the sensors which is the optimal arrangement for bearings only sensors. The shortest geodesics occur when the target moves on the line joining the two sensors. This usually where a location cannot be determined but this singularity is avoided in this example because the sensors are arranged above the plane inhabited by the target but it is still the place where the location is most difficult to determine.

An alternative way to numerically compute the geodesics connecting two points on the manifold *M* is using the Fast Marching Method (FMM) [35,36,37]. Since the Fisher–Rao information metric is a Riemannian metric on the Riemannian manifold *M*, one can show that the geodesic distance map *u*, the geodesic distance from the initial point p to a point θ, satisfies the Eikonal equation
(21)$${∥\nabla u\left(\theta \right)∥}_{g_{\theta }^{-1}}=1$$
with initial condition u(p)=0. By using a mesh over the parameter space, the (isotropic or weakly anisotropic) Eikonal Equation (Equation 21), can be solved numerically by Fast Marching. The geodesic is then extracted by integrating numerically the ordinary differential equation
(22)$$\frac{d\gamma (t)}{dt}=-\eta_{t}g_{\theta (t)}^{-1}\nabla u\left(\theta \left(t\right)\right),$$
where $\eta_{t}>0$ is the mesh size. The computational complexity of FMM is O(NlogN), where *N* is the total number of mesh grid points. We perform FMM on a central computational node in this paper. If the computational capability of that node is limited, a parallel version of FMM maybe used instead and we will include it in future work. For Eikonal equations with strong anisotropy, a generalized version of FMM, the Ordered Upwind Method (OUM) [38] is preferred.

Compare Figure 3 with Figure 4 which uses a Fast Marching algorithm to calculate the geodesic distance from the same point.

Figure 5 shows the speed required along a geodesic travelling in the direction of the vector (1,−1). In the case of these bearing-only sensors, the trade-off over speed is between time-on-target and rate of change of relative angle between the sensors and the target. Travelling slowly means more time for the sensors to integrate measurements but the change in angle is slower, resulting in more accurate position estimates. Conversely, faster movement than the geodesic speed results in larger change in angle measurements but less time for measurements again resulting in more accurate measurements. Only at the geodesic speed is the balance reached and the minimum information criterion achieved.

### 5.2. Configuration Metric Calculations

The coordinates z on $S_{\Omega }$ are $\left\{x_{1},y_{1},x_{2},y_{2}\right\}$. The actual positions of the sensor are raised above Ω by a distance $z_{i}$. The sensor management problem amounts to, given some initial configuration, identifying a choice of $\left\{x_{1},y_{1},x_{2},y_{2}\right\}$ for which the determinant of (Equation 14) is maximized, and traversing geodesics $\gamma^{i}$ in $S_{\Omega }$, starting at the initial locations and ending at the D-optimal locations; see Figure 2. We assume that the target location is given by an uninformative prior distribution; that is, P(θ∈A)=vol(A)/vol(Ω) for all A⊂Ω.

To make the problem tractable, we consider a simple case where one of the sensor trajectories is fixed; that is, we consider a two-dimensional submanifold of M(Γ) parameterized by the coordinates $\left(x_{2},y_{2}\right)$ of $S_{2}$ only. Figure 6 shows a contour plot of det(G), as a function of $\left(x_{2},y_{2}\right)$, for the case where $S_{1}$ moves from (0,1) (yellow dot) to (2,3) (black dot). The second sensor $S_{2}$ begins at the point (−6,−7) (red dot). Figure 6 demonstrates that, provided $S_{1}$ moves from (0,1) to (2,3), det(G) is maximized at the point (−1,−5.5), implying that $S_{2}$ moving to (−1,−5.5) is the D-optimal choice. The geodesic path $\gamma_{1}$ beginning at (0,1) and ending at (2,3) is shown by the dashed yellow curve; this is the information-maximizing path of $S_{1}$ through Ω. Similarly for $S_{2}$, the geodesic path $\gamma^{2}$ beginning at (1,−3) and ending at (−1,−5.5) is shown by the dotted red curve.

### 5.3. Visualizing Configuration Geodesics

Figure 7 shows solutions to the geodesic equation on Ω=[−10,10]×[−10,10] for a sensor starting at (0,1) with the other sensor stationary at (−7,−6) and the target at (−1,−3). The actual positions of the sensor are raised above Ω by a distance 1. The differing paths correspond to ϕ varying through [0,2π] in steps of 0.25 radians in the target initial direction vector (cosϕ,sinϕ). the integration of the geodesic equation is carried to t=5 so the length of each line corresponds to the information obtained about the target. It is clear by that the longer lines correspond to the sensors forming a right-angle relationship to the target which is optimal for bearings-only sensors. This should be compared with Figure 8, which uses a Fast Marching algorithm [39,40] to calculate the geodesic distance from the same point. Figure 9 shows the speed required along a geodesic travelling through each point in the direction of the vector (1,1).

## 6. Discussion

We consider the problem of sensor management from an information-geometric standpoint [8,9]. A physical space houses a target, with an unknown position, and a collection of mobile sensors, each of which takes measurements with the aim of gaining information about target location [12,13,14]. The measurement process is parametrized by the relative positions of the sensors. For example, if one considers an array of sensors that take bearings-only measurements to the target (Equation 18), the amount of information that can be extracted regarding target location clearly depends on the angles between the sensors. In general, we illustrate that in order to optimize the amount of information the sensors can obtain about the target, the sensors should move to positions which maximize the norm of the volume form (‘D-optimality’) on a particular manifold imbued with a metric (Equation 14) which measures the distance (information content difference) between Fisher matrices [30,31]. We also show that, if the sensors move along geodesics (with respect to (Equation 14)) to reach the optimal configuration, the amount of information that they *give away* to the target is minimized. This paves the way for (future) discussions about game-theoretic scenarios where both the target and the sensors are competitively trying to acquire information about one another from stochastic measurements; see, e.g., [41,42] for a discussion on such games. Differential games along these lines will be addressed in forthcoming work.

It is important to note that while the measurements from sensors may be discrete and subject to sampling limits, the underlying model of target and sensor parameters is continuous. We have introduced two discretisations of our manifold to calculate approximations to our optimal reconfiguration paths and to enable some helpful visualisations. The explicit solution of the geodesic equation Figure 3 uses a numerical differential equation solver on a coordinate patch over the sensor manifold. The Fast Marching visualisation of Figure 4 uses a meshgrid over the same coordinate patch. Future work on optimal control of sensor parameters and game theoretic competitions between the information demands of sensors and targets may require more sophisticated discretisations, which available in the literature [43,44].

We hope that this work may eventually have realistic applications to signal processing problems involving parameter estimation using sensors. We have demonstrated that there is a theoretical way of choosing sensor positions, velocities, and possibly other parameters in an optimal manner so that the maximum amount of useful data can be harvested from a sequence of measurements taken by the sensors. For example, with sensors that take continuous or discrete measurements, this potentially allows one to design a system that minimizes the expected amount of time taken to localize (with some given precision) the position of a target. If the sensors move along paths that are geodesic with respect to (Equation 14), then the target, in some sense, learns the least about its trackers. This allows the sensors to prevent either intentional or unintentional evasive manoeuvres; a unique aspect of information-geometric considerations. Ultimately, these ideas may lead to improvements on search or tracking strategies available in the literature, e.g., [45,46]. Though we have only considered simple sensor models in this paper, the machinery can, in principle, be adopted to systems of arbitrary complexity. It would certainly be worth testing the theoretical ideas presented in this paper experimentally using various sensor setups.

## Figures and Tables

**Figure 1 sensors-21-05265-f001:**
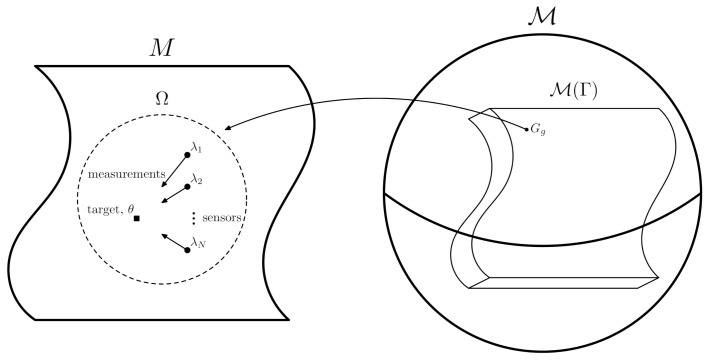
Diagrammatic representation of the sensor model; sensors are at $\lambda_{i}\in M$ taking measurements of a target at θ∈M. A measure of distance between different sensor configurations, physically corresponding to change in information content is obtained through a suitable restriction of the metric $G_{g}$ (Equation 13) to the configuration manifold M(Γ)⊂M, the space of all Riemannian metric on *M*. M is almost certainly not topologically spherical, it is merely drawn here as such for simplicity.

**Figure 2 sensors-21-05265-f002:**
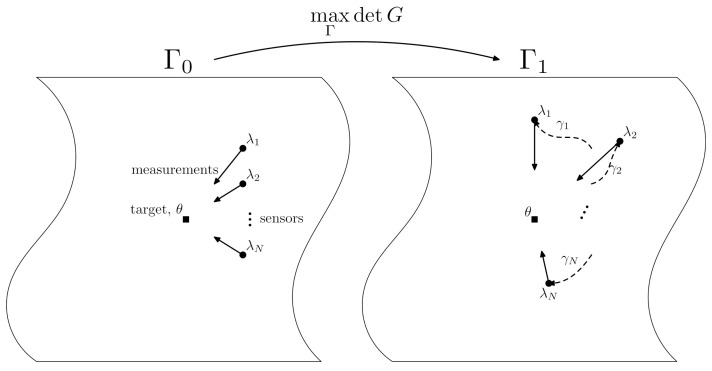
Graphical illustration of D-optimal sensor dynamics; a sensor configuration $\Gamma_{0}$ evolves to a new configuration $\Gamma_{1}$ by moving the *N* sensors in Ω-space to new positions that are determined by maximizing the determinant of the metric *G*, given by Equation (Equation 14), on the sensor manifold M(Γ). Each sensor $\lambda^{i}$ traverses a path $\gamma_{i}$ through Ω-space to end up in the appropriate positions constituting $\Gamma_{1}$. As shown in Section 4.3, the paths $\gamma_{i}$ are entropy minimizing if they are geodesic on the sensor manifold M(Γ). Note that the target is shown as stationary in this particular illustration.

**Figure 3 sensors-21-05265-f003:**
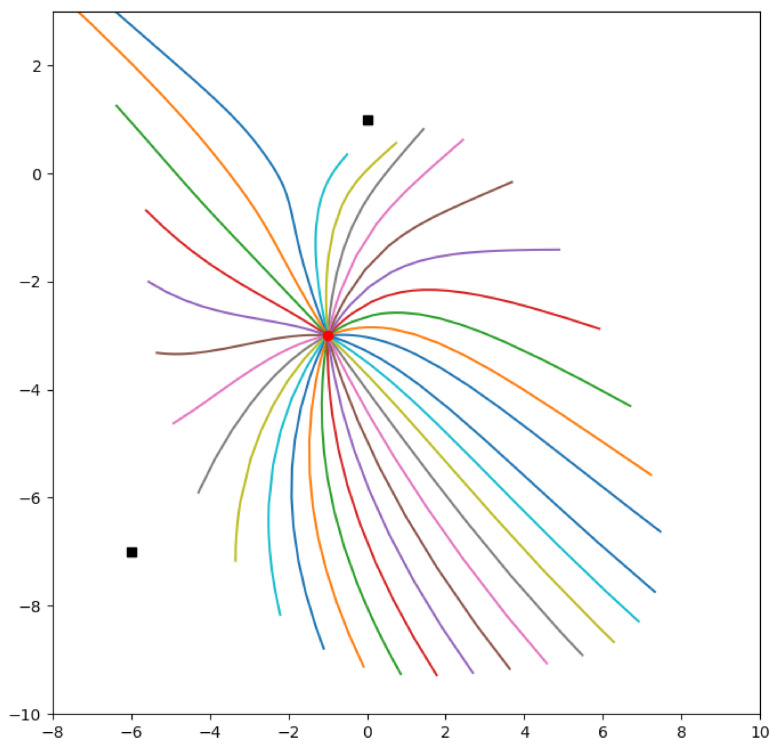
Solutions to the geodesic equation on Ω=[−10,10]×[−10,10] for a target starting at (−1,−3) and sensors at (−7,−6) and (0,1) (raised at a height 1 above Ω). The differing paths correspond to the initial direction vector (cosθ,sinθ) of the target, varying as θ varies from 0 to 2π radians in steps of 0.25 radians.

**Figure 4 sensors-21-05265-f004:**
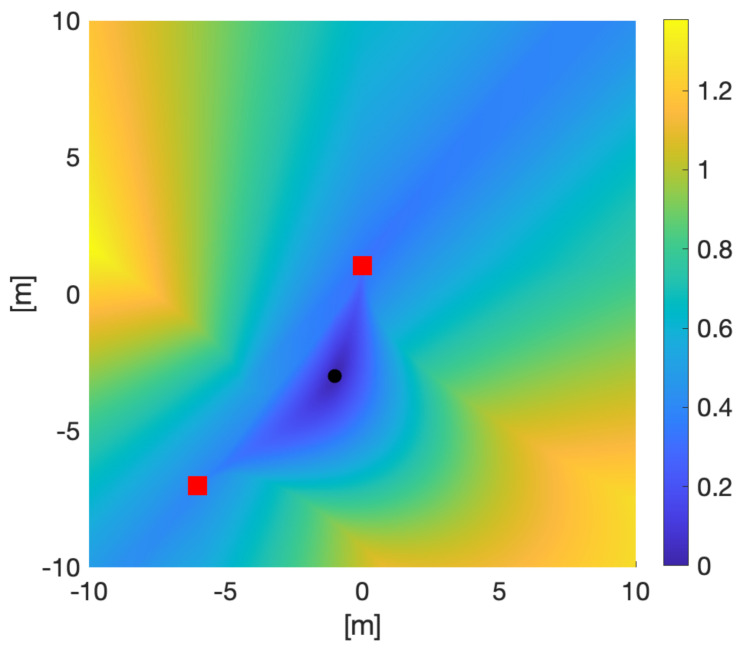
Geodesic distance on Ω=[−10,10]×[−10,10] from the point (−1,−3) with sensors at (−7,−6) and (0,1) (raised at a height 1 above Ω). The distance was calculated using a Fast Marching formulation of the geodesic equation. The fact that the geodesics follow the gradient of this distance allows comparison with Figure 3.

**Figure 5 sensors-21-05265-f005:**
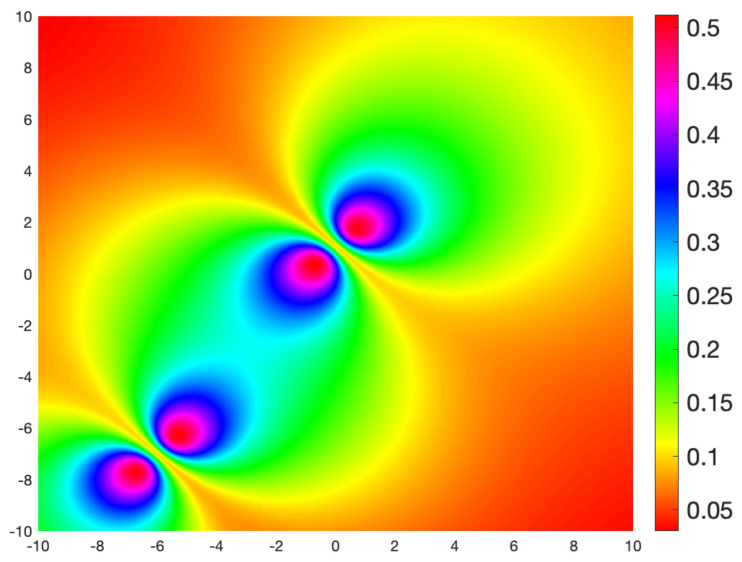
Geodesic speed for each point in Ω=[−10,10]×[−10,10] for targets departing each point in direction of the vector (1,−1).

**Figure 6 sensors-21-05265-f006:**
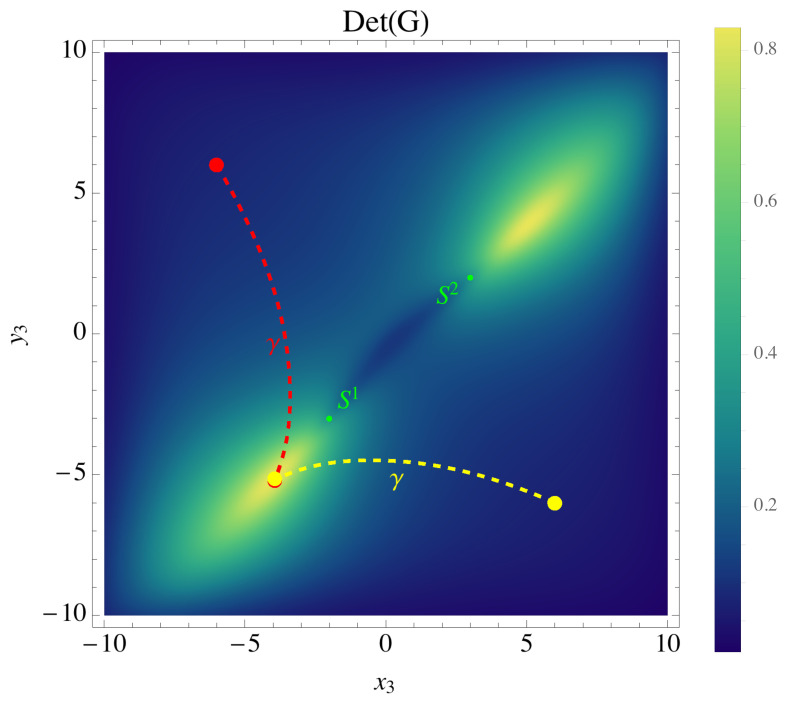
Contour plot of det(G), for G given by (Equation 14), for the case of three sensors, where $S^{1}$ is fixed at (−2,−2), and $S^{2}$ at (2,2) (red dot). Brighter shades indicate a greater value of det(G); det(G) is maximum at (−1,−5.5). The third sensor is allowed to move. The geodesic path linking the initial and final positions of $S^{3}$ starting at (6,−6) (yellow dot) is shown by the dashed yellow curve, while the dashed red curve shows another geodesic path linking (−6,−7) (red dot) to the D-optimal location (−1,−5.5). The actual positions of the sensors are raised above Ω by a distance $z_{i}$.

**Figure 7 sensors-21-05265-f007:**
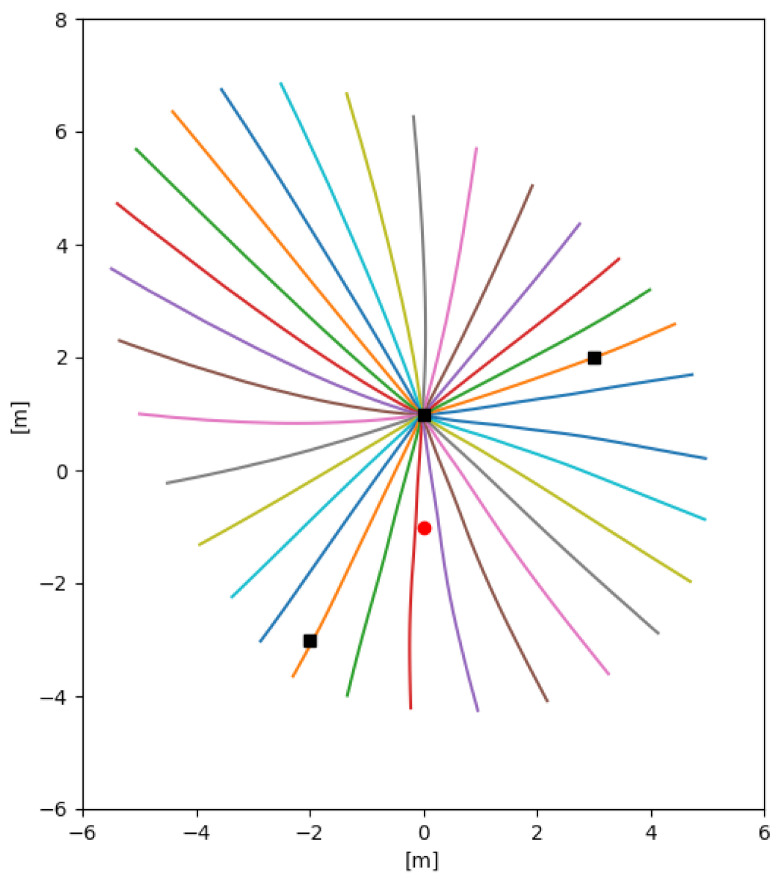
Solutions to the geodesic equation on Ω=[−10,10]×[−10,10] for a sensor starting at (0,1) with the other sensors stationary at (−2,−3) and (3,2) (raised at a height 1 above Ω.). The target is located at (0,1) on Ω. The differing paths correspond to the initial direction vector (cosϕ,sinϕ) of the target, varying as ϕ varies from 0 to 2π radians in steps of 0.25 radians.

**Figure 8 sensors-21-05265-f008:**
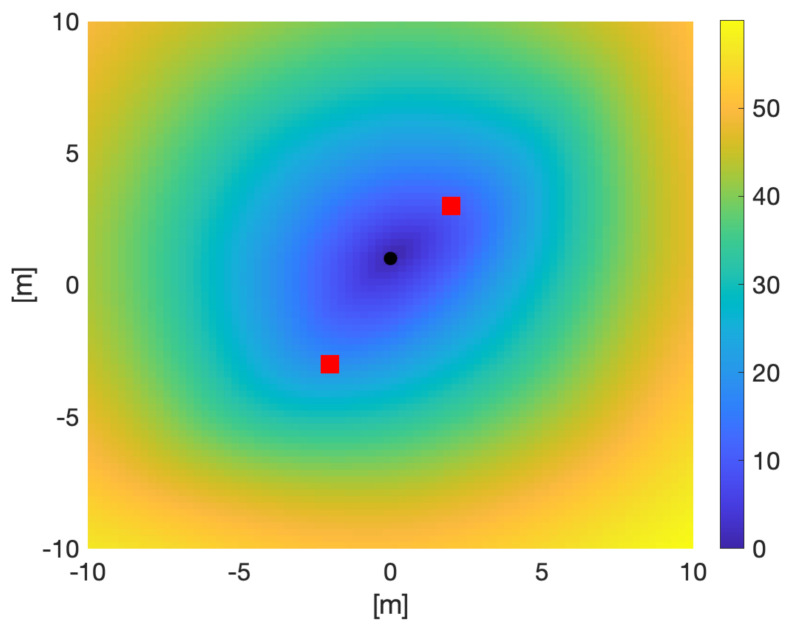
Geodesic distance on Ω=[−10,10]×[−10,10] for a sensor starting at (0,1) with the other sensor stationary at (−7,−6) (raised at a height 1 above Ω) and the using an uninformative prior for the target. The distance was calculated using a Fast Marching formulation of the geodesic equation. The fact that the geodesics follow the gradient of this distrance allows comparison with Figure 7.

**Figure 9 sensors-21-05265-f009:**
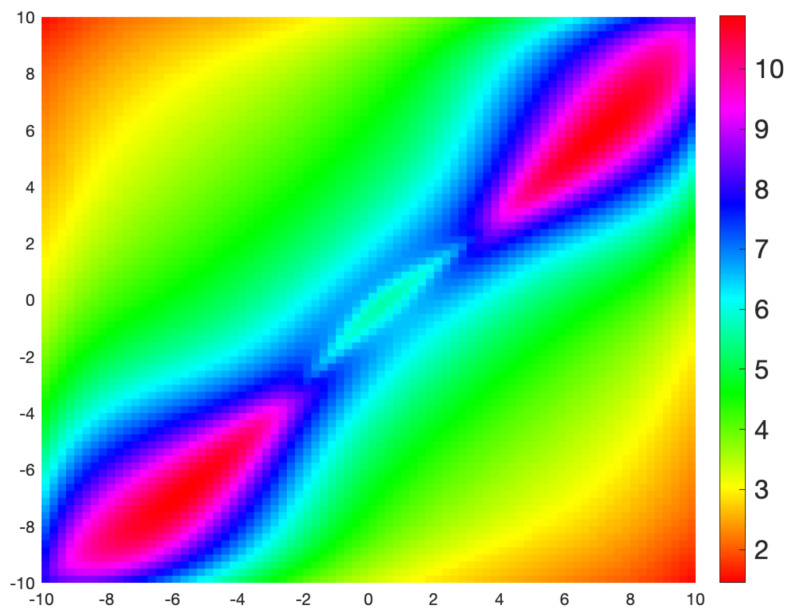
Geodesic speed sensor moving along a geodesic in direction (1, −1) at each point in Ω=[−10,10]×[−10,10]. The other sensor is stationary at (−7,−6), and an uninformative prior is used for the target. The actual positions of the sensor are raised above Ω by a distance 1.

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
