# Peer review of "The Information Geometry of Sensor Configuration"

_sensors, 2021, doi:10.3390/s21165265_

Round 1

Reviewer 1 Report

The manuscript addresses the important practical problem of sensor management from an information-geometric point of view. The authors face the problem of parameter estimation from sensor data and give many valuable conclusions and remarks taking into account that a change in the sensor parameters results in a corresponding change to the metric. The paper is professionally written and well structured. It provides accurate and up-to-date literature review. A proof of presented theorem is given in the supplementary material in detail. An example illustrating the proposed mathematical model and its application is theoretical, however the authors indicate a number of practical applications of the proposed approach, which in my opinion is sufficient and the manuscript can be accepted in its present form. The only remark is technical: the Figures can be given in the paper close to the text in which they are discussed.

Author Response

We shall move the figures closer to their mention in the text

Reviewer 2 Report

The paper is well written and technically correct.

Author Response

We thank the reviewer for their kind comments

Reviewer 3 Report

For the future work, and for this one,  it will be useful to explain for the reader how the descriptions can be used in applications and way this solution is better than others from existing state of art research.

A comparison with other existing solution could be useful.

Author Response

We respect and understand the editors and reviewers demand for a
  comparison with an alternative approach but this is, by and large, a theory paper that sets up an approach to the understanding of 
sensors in a generic sense.  We hope that we have advanced the theory of sensing in that way, and indeed believe that 
our results are interesting in that respect. 

It is not a "methods" paper. The scenario we discussed  was an idealised one and, as we hope we indicated,  there to assist in the reader's understanding of the theory. Likewise, the purpose of our simulations  was not to show how good our approach was, but  to demonstrate the shapes of the optimal paths (geodesics) and thereby   provide further  assistance to the understanding of  the reader.
We are struggling with the idea of comparison with another "method" in this context. Could the editor/reviewers give some advice about what kind of comparison they would like?

In ``An approach to the selection of optimal sensor locations in distributed parameter systems'' by 
Alain Vande Wouwer, Nicolas Point, Steaphanie Porteman, Marcel Remy (Journal of Process Control 10 (2000) 291--300) the optimality criterion is a measure of  independence of the measurements. But, this is a relative optimality dependent on the particular set of measurements, as distinct from our approach which delineates the trajectory of sensor parameters the generates the maximal information increase over all possible measurements.